# Characteristics of hospital pediatricians and obstetricians/gynecologists working long hours in Tokushima, Japan: A cross-sectional study

**Mai Nakagawa[1,2], Kazumi Nakagawa[1], Koga Nakai[1], Ayumu Tominaga[1], Yoshiro Mori[1], Takeshi Iwasa[3], Maki Urushihara[4], Ichiro Hashimoto[2], Hisayoshi Morioka[1]***

1 Department of Public Health, Graduate School of Biomedical Sciences, Tokushima University, Tokushima city, Tokushima, Japan, 2 Department of Plastic and Reconstructive Surgery, Graduate School of Biomedical Sciences, Tokushima University, Tokushima city, Tokushima, Japan, 3 Department of Obstetrics and Gynecology, Graduate School of Biomedical Sciences, Tokushima University, Tokushima city, Tokushima, Japan, 4 Department of Pediatrics, Graduate School of Biomedical Sciences, Tokushima University, Tokushima city, Tokushima, Japan

* hisayoshi.morioka@tokushima-u.ac.jp

## Abstract

### Background

This study aimed to determine the actual working conditions, including working hours and desired future working styles of hospital pediatricians and obstetricians/gynecologists (OB/GYNs) in Tokushima Prefecture.

### Method

This cross-sectional study used a self-administered questionnaire. Pediatricians and OB/GYNs ($n = 96$) working at 14 hospitals in Tokushima Prefecture were surveyed. The questionnaire included items related to working hours, working status, number of medical institutions, task-shifting/task-sharing status, and preferred work style. Factors associated with work hours were analyzed.

### Results

Approximately 40% and 10% of pediatricians and OB/GYNs worked ≥60 h/week and ≥80 h/week, respectively. Their weekly working hours were significantly positively associated with the number of nights and holidays worked and concurrent medical facilities they worked at. Multiple regression models showed that weekly working hours were significantly associated with working at night and day-off duties and the number of working medical institutions being worked at concurrently. Pediatricians and OB/GYNs who worked ≥60 h/week were not implementing a "multiple attending physician system" or "on-call system," and task-shifting/task-sharing was inadequate. A significantly higher proportion of physicians who worked long hours (≥60 h/week or ≥80 h/week) wanted to reduce the number of night and day-off duties and work hours compared with those who did not.

**Data Availability Statement:** All relevant data are within the manuscript and its Supporting Information files.

**Funding:** This study was financially supported by Tokushima Prefecture [www.pref.tokushima.lg.jp] in the form of a grant received by HM. No additional external funding was received for this study.

**Competing interests:** The authors have declared that no competing interests exist.

**Abbreviations:** OB/GYNs, obstetricians/ gynecologists; OB/GYN, obstetrics/gynecology; NICU, neonatal intensive care unit; MFICU, maternal-fetal intensive-care unit; COVID-19, coronavirus disease 2019.

## Conclusions

The results of this study suggest that many pediatricians and OB/GYNs work long hours, and it is essential to improve their work environment, including task-shifting/task-sharing. Further enhanced recruitment and retention of hospital pediatricians and OB/GYNs is required. A detailed and large-scale study of pediatricians and OB/GYNs' working environments is essential in the future.

## 1 Background

Working hours for workers including physicians are regulated by the law in each country. For example, the workweek is 40 h in Japan [1], the average per week for 17 weeks is 48 h in the UK [2], and it is 8 h per day and 6 days per week in Germany [3]. However, there are exceptions, including exemptions and special exceptions, which allow workers to work hours exceeding the statutory working hours. Working hours vary widely depending on the nature and amount of work. Therefore, examining the nature and quantity of work as well as to determine the hours worked for each type of job in each region is important.

Long working hours are associated with an increased risk of adverse physical health outcomes, such as coronary heart disease [4, 5], stroke [5], and diabetes [6], and with deleterious effects on mental health, such as depression [6] and suicide [7]. Additionally, regarding occupational safety, long working hours are reportedly associated with near-misses (that is, "an unplanned event that did not result in injury, illness, or damage but had the potential to do so") and injuries [8]. Long working hours are a public health concern that must be addressed.

Burnout among physicians has recently become a global concern [9–11]. Burnout comprised three symptoms: emotional exhaustion, depersonalization, and low personal accomplishment [12]. Moreover, one of the factors associated with burnout is working long hours [13–15]. In a large-scale Japanese study, approximately 40% and 10% of physicians worked >60 h/week and >80 h/week, respectively [16]. In another Japanese study, full-time hospital physicians worked 50.1%/week [17]. In addition to them, 10.5% of full-time physicians aged 24–69 years worked more than 80 h/month overtime in the main hospital and 4.4% worked ≥80 h/month overtime in side work [18]. In Taiwan, approximately 35% of physicians worked >65 h/week [19]. Notably, in a representative US study, 58.9% of surgical residents worked >80 h/week [20]. Additionally, in a representative Japanese study, 20.1% of postgraduate residents worked >80 h/week [21]. In another Japanese study, 67% and 27% of resident physicians worked ≥60 and ≥80 hours /week in their hospitals respectively [22]. However, reports on the working hours of pediatricians and obstetricians/gynecologists (OB/GYNs) are limited.

Tsutsumi suggested that doctor shortages and their uneven distribution between regions and specialties are related to the long doctors' working hours in addition to a lack of task-sharing [23]. In Japan, pediatricians [24–26] and OB/GYNs [27, 28] are unevenly distributed among medical specialties. Nomura et al. reported that many rural hospitals in Japan have closed their pediatric and OB/GYN departments [25]. By correcting long working hours, rural hospitals would have sufficient pediatricians and OB/GYNs. One possible solution is improving hospital and regional retention of physicians. One of the factors associated with regional retention of physicians is the training environment and career support [29, 30]. If the characteristics of pediatricians and OB/GYNs' long working hours in hospitals are identified, they can be reduced through effective collaboration between the hospital staff and task-sharing.

In Japan, exceeding the upper limit of working hours was determined by each institution, such as hospitals, based on a labor-management agreement. Previously, there was no

uniformity in the overtime limits for workers including physicians based on the law. However, the Labor Standards Act will be revised to apply overtime regulations to physicians starting in the fiscal year (FY) 2024. The overtime limits for resident doctors and physicians working in emergency departments are 1,860 h (equivalent to an 80-h work/week) and 960 h (equivalent to a 60-h work/week), respectively [31]. Pediatricians and OB/GYNs are responsible for policy-based medicine, such as pediatric emergencies and perinatal and neonatal medicine. Compliance with the law may affect local medical care systems. Therefore, it is necessary first to estimate hospital pediatricians and OB/GYNs' working hours, including side work hours. Hence, this study aimed to determine the actual working conditions, including working hours and desired future work styles of hospital pediatricians and OB/GYNs in Tokushima Prefecture, Japan, and determine the characteristics of those working long hours.

## 2 Methods

### 2.1 Ethics of the research

This study was conducted in accordance with the Declaration of Helsinki and National Ethical Guidelines. Written informed consent was obtained from all the participants. The study protocol was approved by the Ethics Committee of Tokushima University Hospital (approval number: 4077, approval date: 27 September 2021, reference number of ethics committee: 11000161).

### 2.2 Study aim, design, and setting

This study aimed to determine the actual working conditions, including working hours, and desired future work styles of hospital pediatricians and OB/GYNs in Tokushima Prefecture. This cross-sectional study was conducted in the Tokushima Prefecture, Japan.

### 2.3 Study participants

This study included pediatricians and OB/GYNs working as full staff in 14 hospitals in Tokushima Prefecture, Japan. The Ministry of Health, Labor, and Welfare's study of physicians, dentists, and pharmacists conducted every 2 years reported that in 2020, 62 pediatricians and 57 OB/GYNs were working in hospitals in the Tokushima Prefecture.

A letter explaining the study, self-administered questionnaire, and return envelope were mailed to each physician supervising pediatrics and obstetrics/gynecology departments at 14 hospitals. The supervising physicians distributed these forms to physicians working in the respective hospital departments. The cover page of the study form stated that participation is voluntary and that no personally identifiable data will be provided to the medical institutions where they work. Moreover, the study form included a box to confirm participants' informed consent. An identification number was assigned to each hospital, and a participant's name was not required. Each physician who gave consent completed the questionnaire and returned it in a sealed envelope. No hospital's administrative office permission was needed to conduct this survey. Between 1 October 2021 and 31 January 2022, 96 participants were included in the analysis. The participants' characteristics are listed in Table 1.

### 2.4 Measurements

The questionnaire, supporting information1, included items related to the physicians' age, sex, specialty, and type of practice at their workplace: day shift work- and night and holiday work-arrangements. The former included "primary attending physician system," "multiple attending physician system," and "others (for example, working in neonatal intensive care unit [NICU]/

**Table 1. Characteristics of the analyzed participants.**

|  | *n* | % |
|---|---|---|
| **Sex** |  |  |
| Male | 48 | 50.0 |
| Female | 48 | 50.0 |
| **Age** |  |  |
| <30 years | 8 | 8.3 |
| 30–39 years | 28 | 29.2 |
| 40–49 years | 28 | 29.2 |
| 50–59 years | 18 | 18.8 |
| 60–69 years | 11 | 11.5 |
| ≥70 years | 3 | 3.1 |
| **Specialty** |  |  |
| Pediatrics | 50 | 52.1 |
| Obstetrics/Gynecology | 45 | 46.9 |
| Others | 1 | 1.0 |
| **Working hours (per week)** |  |  |
| <60 hours | 55 | 57.3 |
| 60–79 hours | 29 | 30.2 |
| ≥80 hours | 12 | 12.5 |
| **Number of medical institutions they work at concurrently** |  |  |
| 1 institution | 53 | 55.2 |
| 2–3 institutions | 25 | 26.0 |
| ≥4 institutions | 18 | 18.8 |
| **Number of night and dayoff duties (per week)** |  |  |
| <once/week | 33 | 34 |
| ≥once/week, <twice/week | 27 | 28.1 |
| ≥twice/week | 36 | 37.5 |
| **Number of annual paid leave days (last year)** |  |  |
| <5 days/year | 18 | 18.8 |
| 5–9 days/year | 45 | 46.9 |
| ≥10 days/year | 26 | 27.1 |
| Unknown | 7 | 7.3 |
| **Characteristics of daytime working status** |  |  |
| One attending physician system (per some patients) | 37 | 38.5 |
| Multiple attending physician system (per some patients) | 50 | 52.1 |
| Others (including NICU/MFICU) | 9 | 9.4 |
| **Characteristics of nighttime working status** |  |  |
| Shift work system | 24 | 25.0 |
| On-call system | 51 | 53.1 |
| Others (including NICU/MFICU.) | 21 | 21.9 |
| **Sharing work on measuring vital signs** |  |  |
| Always | 82 | 85.4 |
| Not always | 14 | 14.5 |
| **Sharing work on inputting medical records** |  |  |
| Always | 44 | 45.8 |
| Not always | 52 | 54.2 |
| **Sharing work on writing documents (including certificates)** |  |  |
| Always | 51 | 53.1 |

(*Continued*)

**Table 1.** (Continued)

|  | *n* | % |
|---|---|---|
| Not always | 45 | 46.9 |
| **Sharing work on transporting patients and carrying luggage** |  |  |
| Always | 71 | 74.0 |
| Not always | 25 | 26.0 |
| **Desire to decrease days of night duties** |  |  |
| Yes | 28 | 29.2 |
| No | 68 | 70.8 |
| **Desire to decrease days of day-off duties** |  |  |
| Yes | 17 | 17.7 |
| No | 79 | 82.3 |
| **Desire to decrease days of overtime working hours** |  |  |
| Yes | 37 | 38.5 |
| No | 59 | 61.5 |

Analyzed participants: *n* = 96

Abbreviations: NICU, neonatal intensive care unit; MFICU, maternal-fetal intensive-care unit.

maternal-fetal intensive-care unit [MFICU]),” while the latter included “on-call system,” “shift work system,” and “others (for example, working in NICU/MFICU).”

The following items are related to their work status:

Number of medical institutions where the respondents worked in September 2021; average hours worked per week (medical activities only, excluding research, training, or teaching activities, and if the respondent works at more than one medical institution, the total number of hours worked per week was calculated for the entire month); number of times per month that the participant worked night and day-off duties (at all medical institutions where they worked); number of annual paid leave days in 2020; status of task-shifting/task-sharing between physicians and nonphysicians; and desired future work style.

The following six items were included based on previous studies on task-shifting/task-sharing status between physicians and nonphysicians [16].

“Explanation and consensus building with patients,” “Taking basic vitals, such as blood pressure,” “Simple procedures to securing an intravenous line for intravenous infusion, taking blood samples, and data acquisition,” “Inputting medical records (electronic medical record entry),” “Medical clerical work (preparation of medical certificates and other documents, such as patient appointments),” and “Transporting and restocking supplies in the hospital and transporting patients to and from laboratories.” Three responses were used for each item: “I can always share” (considered as “Always”), “I can sometimes share,” and “I cannot share at all” (considered as “Not always”). The following four items were included in the desired future work style: Decrease in overtime work hours per week, number of day-off duty shifts per month, number of night duty shifts per month, and number of on-calls per month. The respondents were asked to answer “yes” if their current working style was applicable and “no” if not.

## 2.5 Statistical analysis

Total working hours were classified into <60, 60–80, and ≥80 h/week. The 60 h/week and 80 h/week working hours are equivalent to 1,000 and 2,000 h, respectively, as annual overtime levels, given that the legal working hours stipulated by the Japanese Labor Standards Act is 40

h/week. In Japan, the Labor Standards Act will be revised to apply overtime regulations to physicians starting in the fiscal year (FY) 2024. The overtime limits for resident doctors and physicians working in emergency departments are 1,860 h (equivalent to an 80-h work/week) and 960 h (equivalent to a 60-h work/week), respectively [31]. This category can be used to evaluate the portions of the overtime regulations that need to be addressed.

First, the number of pediatricians and OB/GYNs working ≥60 h/week and ≥80 h/week was calculated based on age group.

Second, we calculated the mean and standard deviation (SD) of their age by sex, weekly working hours, number of night and day-off duties, number of medical institutions being worked at concurrently, and number of annual paid leave days. Sex differences in the means of those continuous variables were analyzed using Student's t-test.

Third, Pearson's Correlation Coefficient was used to evaluate the relationship between the weekly working hours and other variables. Subsequently, we analyzed the comparative effects on weekly working hours using hierarchical regression. In Model 1, the regression model was populated with working at night and day-off duties as independent variables. In Model 2, the number of medical institutions being worked at concurrently was added to the independent variable in Model 1. In Model 3, the number of annual paid leave days was added to the independent variable in Model 2. Adjusted R-squared values were calculated to check the degree of deviation of each model. The multicollinearity of independent variables was examined using the variation inflation factor (VIF). Independent variables indicating 10 ≥VIF were assumed to be multicollinear. The VIF of all independent variables was less than 1.4.

Forth, the chi-square test was used to analyze the differences in day and nighttime working status and work-sharing status between pediatricians and OB/GYNs who worked long hours (≥60 h/week or ≥80 h/week). Finally, we analyzed the differences in the desired working environment between pediatricians and OB/GYNs using the chi-square test, dividing the working hours into <60 h/week, ≥60 h/week, <80 h/week, and ≥80 h/week.

The statistical tests used are listed in the legend of Tables and Figures. Statistical tests were based on two-sided probabilities, and a $p$-value <0.05 was considered significant. All statistical analyses were performed using IBM SPSS Statistics version 28.0 for Windows (IBM; Armonk, NY, USA).

## 3 Results

The participants' characteristics are presented in Table 1 ($n$ = 96). The number of pediatricians and OB/GYNs working ≥60 h/week are shown in Fig 1(A). Fig 1(B) shows the number of pediatricians and OB/GYNs working for ≥80 h/week. The number of pediatricians and OB/GYNs working ≥60 h/week was the largest at 12 in the 30–39 y.o. and 50–59 y.o. groups. The breakdown was six pediatricians and obstetricians in the 30–39 y.o. group, and five pediatricians and seven obstetricians in the 50–59 y.o. group. The number of pediatricians and OB/GYNs working ≥80 h/week was the largest at four in the 50–59 y.o. group.

The participants' characteristics of gender difference are presented in Table 2 ($n$ = 96). The mean and SD of the age for males and females were 49.5 (13.3) and 40.8 (7.7), respectively. The mean age of men was significantly higher than that of women ($p$<0.001). The mean age and standard deviation were 49.5 (13.3) years for men and 40.8 (7.7) years for females, respectively. Men worked longer hours than women; however, the difference was not significant ($p$ = 0.115). The mean and SD of the number of night and day-off duties (per week) were 6.5 (4.3) times for men and 4.8 (3.9) times for women, respectively. Males were significantly more frequent than females ($p$ = 0.042). The mean and SD of the number of medical institutions concurrently being worked at were 2.1 (1.6) for males and 2.0 (1.5) for females, respectively.

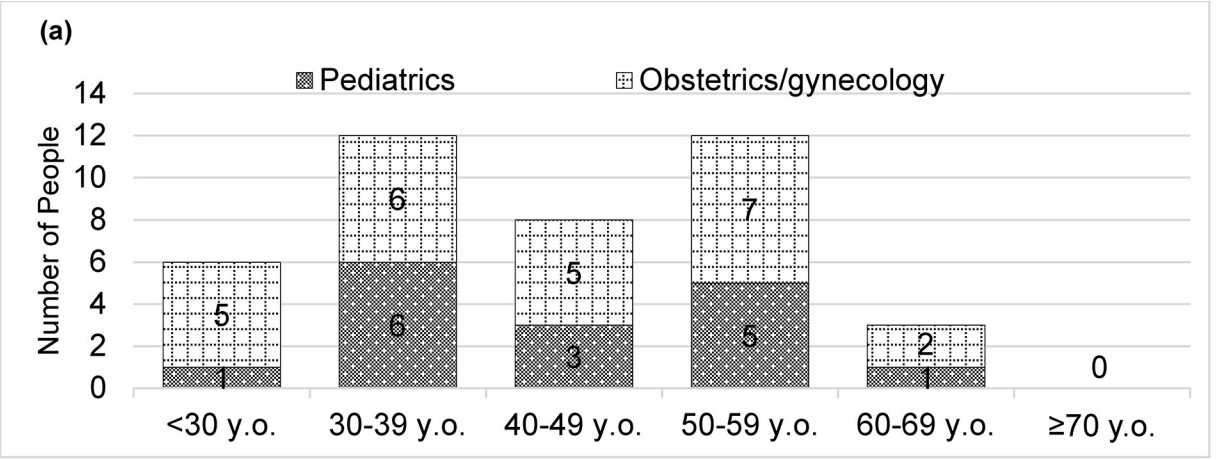

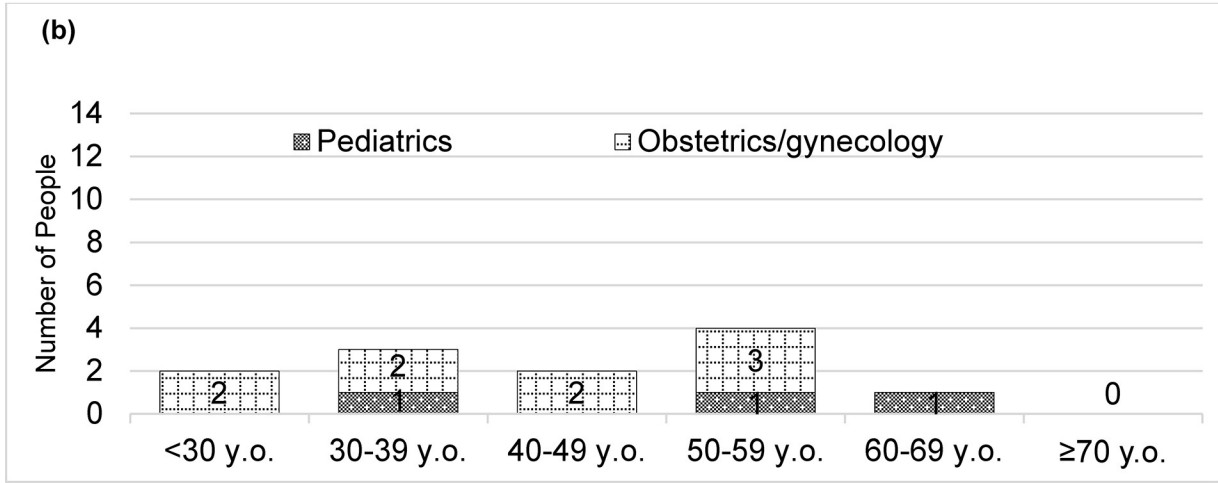

**Fig 1. Numbers of pediatricians/obstetricians working long hours, based on age.** (a) Working ≥60 hours ($n$ = 95). (b) Working ≥80 hours ($n$ = 95). Participants with missing data were excluded from the analysis.

Males worked in more medical institutions concurrently than females; however, the difference was not significant ($p$ = 0.235). The mean and SD of the number of annual paid leave days (last year) were 6.4 (5.4) for men and 7.5 (5.1) for women, respectively. Women had more annual paid leave days than men; however, the difference was not significant ($p$ = 0.338).

**Table 2. Characteristics of gender differences of the analyzed participants.**

|  | Male | | Female | | |
|---|---|---|---|---|---|
|  | $n$ | mean (SD) | $n$ | mean (SD) | $p$ value |
| Age (years) | 48 | 49.5(13.3) | 48 | 40.8(7.7) | <0.001 |
| Working hours (per week) | 48 | 60.7(21.9) | 48 | 54.4(16.8) | 0.115 |
| Number of night and day-off duties (per week) | 48 | 6.5(4.3) | 48 | 4.8(3.9) | 0.042 |
| Number of medical institutions being worked at concurrently | 48 | 2.2(1.6) | 48 | 2.0(1.5) | 0.235 |
| Number of annual paid leave days (last year) | 46 | 6.4(5.4) | 43 | 7.5(5.1) | 0.338 |

Abbreviations: SD, standard deviation.

**Table 3. Relationships between working hours and the other variables.**

| Variable | Total | 1 | | 2 | | 3 | | 4 | | 5 | |
|---|---|---|---|---|---|---|---|---|---|---|---|
| | n | r | p value | r | p value | r | p value | r | p value | r | p value |
| 1. Working hours (per week) | 96 | 1 | | | | | | | | | |
| 2. Age | 96 | -0.14 | 0.180 | 1 | | | | | | | |
| 3. Working at night and day-off duties | 96 | 0.56 | <0.001 | -0.09 | 0.379 | 1 | | | | | |
| 4. Number of medical institutions being worked at concurrently | 96 | 0.46 | <0.001 | -0.13 | 0.215 | 0.39 | <0.001 | 1 | | | |
| 5. Number of annual paid leave days (last year) | 89 | -0.30 | 0.004 | 0.16 | 0.139 | -0.13 | 0.244 | -0.38 | <0.001 | 1 | |

Abbreviations: r, correlation coefficient.

Participants with missing data were excluded from the analysis.

For the calculation of the p values, Student's t-test was used.

Relationships between the working hours and other variables are presented in Table 3. Working hours (per week) correlated with the number of night and day-off duties (r = 0.56, p<0.001) and medical institutions being worked at concurrently (r = 0.46, p<0.001). Working hours (per week) were negatively correlated with number of annual paid leave days (r = - 0.30, p = 0.004). Furthermore, the number of night and day-off duties positively correlated with the number of medical institutions being worked at concurrently (r = 0.39, p<0.001). Working hours (per week) and age were not significantly associated (r = - 0.14, p = 0.180).

Participants with missing data were excluded from the analysis.

For the calculation of the linear correlation coefficient and p values, Pearson correlation analysis was used.

Associations between the working hours and other variables are presented in Table 4. The hierarchical regression analysis showed that weekly working hours were significantly associated with working at night and day-off duties (β = 0.42, p<0.001) and the number of medical institutions being worked at concurrently (β = 0.26, p = 0.011). Working at night and day-off duties had a greater impact on weekly working hours than the number of working medical institutions being worked at concurrently.

Participants with missing data were excluded from the analysis.

For the calculation of the standardized regression coefficient and p values, hierarchical regression analysis was used to predict weekly working hours.

The percentage of physicians working ≥60 h/week for daytime was lowest in the "multiple attending physician system" (34.0%, p = 0.022) [Fig 2(A)]. The on-call system" had the lowest percentage of physicians working ≥60 h/week at night (25.0%, p = 0.012) [Fig 2(B)].

The percentage of physicians working ≥60 h was significantly lower for work sharing on measuring vital signs, inputting medical records, and writing documents (certificates) than

**Table 4. Associations between the working hours and other variables.**

| Variables (n = 89) | Model 1 | | Model 2 | | Model 3 | |
|---|---|---|---|---|---|---|
| | β | p value | β | p value | β | p value |
| Working at night and day-off duties | 0.54 | <0.001 | 0.42 | <0.001 | 0.42 | <0.001 |
| Number of medical institutions being worked at concurrently | | | 0.32 | <0.001 | 0.26 | 0.011 |
| Number of annual paid leave days (last year) | | | | | -0.15 | 0.097 |
| Adjusted $R^2$ | 0.29 | | 0.36 | | 0.38 | |
| F test (f value, p value) | 36.07 | <0.001 | 26.08 | <0.001 | 18.70 | <0.001 |

Abbreviations: β, standardized regression coefficient.

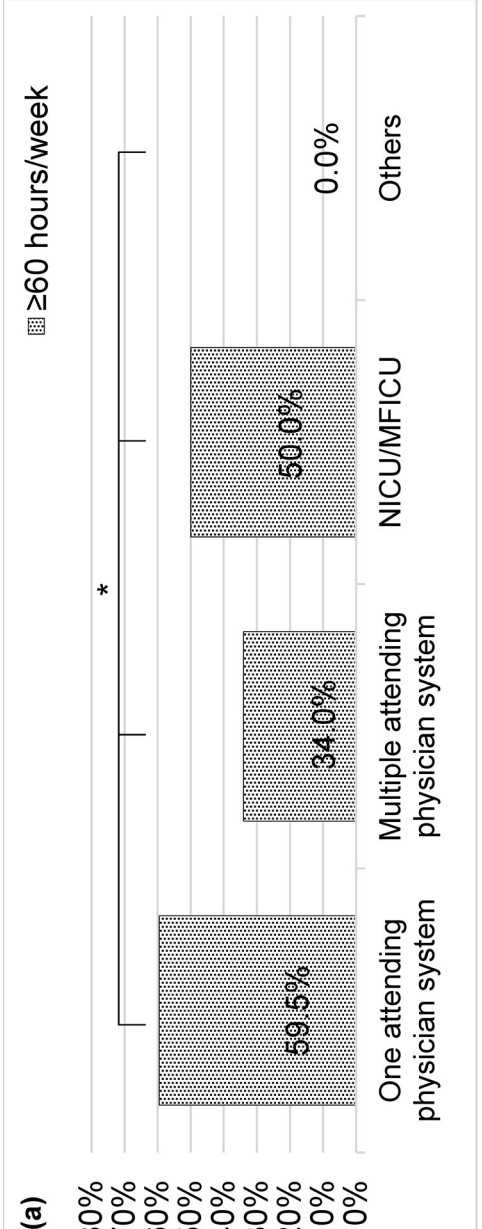
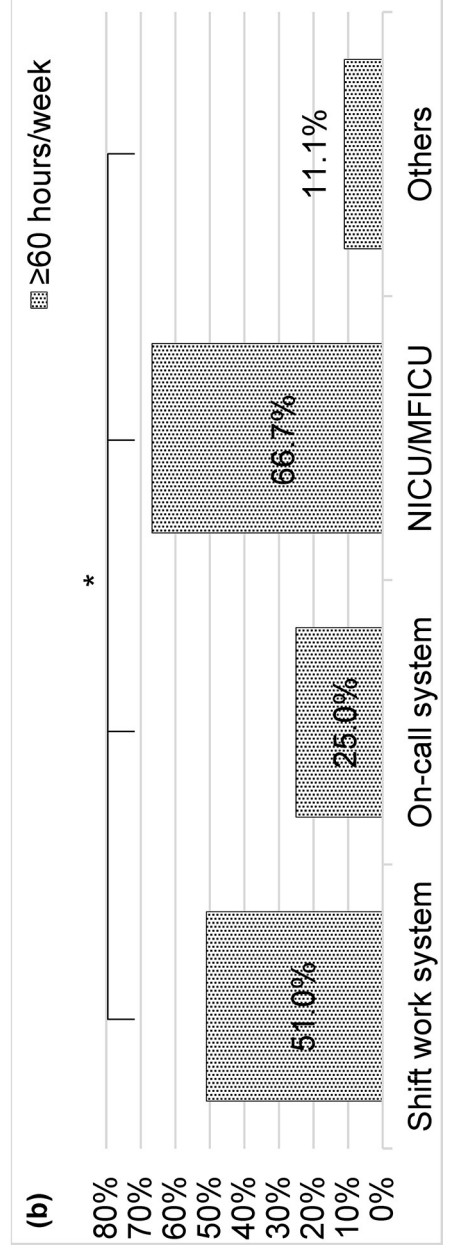

**Fig 2. Percentages of pediatricians and obstetricians/gynecologists working long hours, based on working status.** (a) Daytime working status (*n* = 96, *p* = 0.022). (b) Nighttime working status (*n* = 96, *p* = 0.012). P value was calculated by χ2-test. *$p < 0.05$, ** $p < 0.01$. Abbreviations: NICU, neonatal intensive care unit; MFICU, maternal-fetal intensive-care unit.

those who always can do so but not always able to do so (37.8% vs. 71.4%, *p* = 0.019) [Fig 3 (A)], (29.5% vs. 53.8%, *p* = 0.016) [Fig 3(B)], (31.4% vs. 55.6%, *p* = 0.017) [Fig 3(C)]. The percentage of physicians working ≥80 h was significantly lower in work sharing on transporting patients than those who were always able but not always able to do so (9.8% vs. 28.6%, *p* = 0.043) [Fig 3(D)].

The percentage of physicians who wanted to decrease the number of night duties, day off duties, and overtime work increased significantly as the number of hours worked per week (<60 h/week, ≥60 h/week, <80 h/week, or ≥80 h/week) increased (10.9% vs. 48.3% vs. 66.7%,

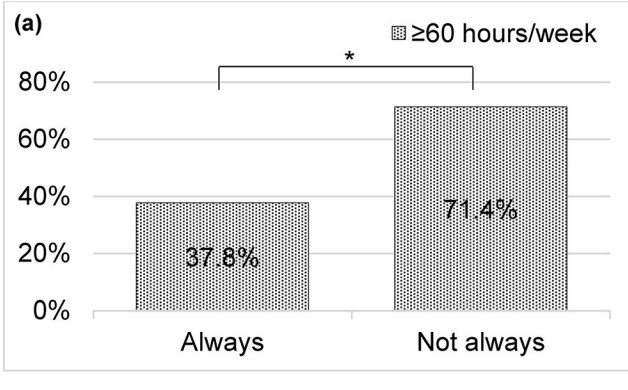
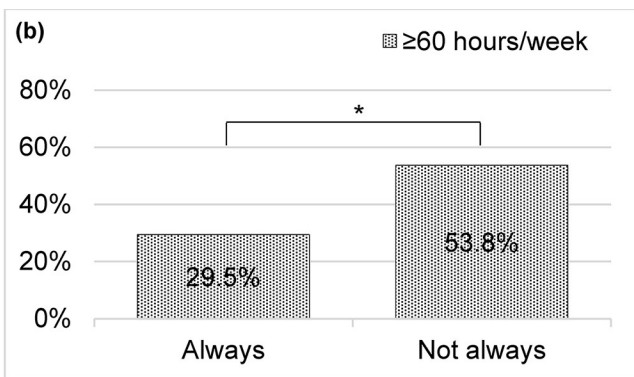
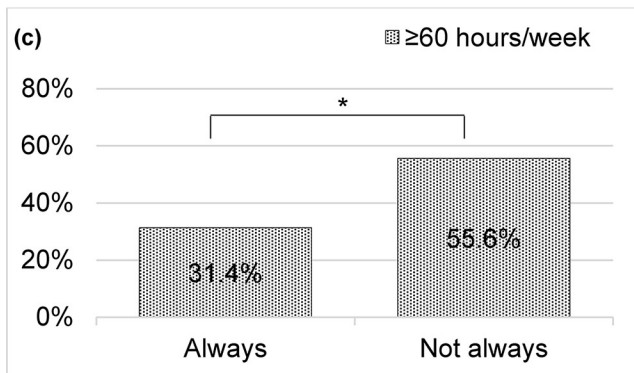
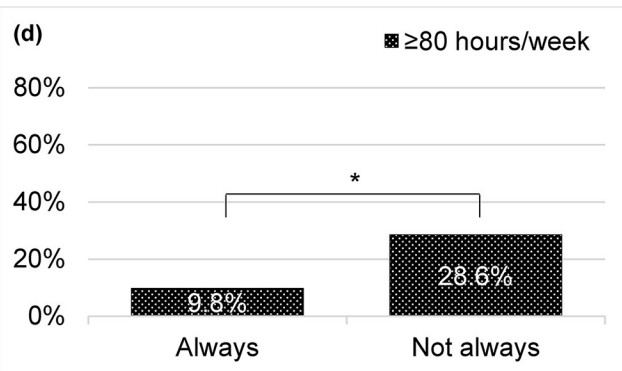

**Fig 3. Percentages of pediatricians and obstetricians/gynecologists working long hours, based on work sharing.** (a) Work sharing on measuring vital signs ($n = 96$, $p = 0.019$). (b) Work sharing on inputting medical records ($n = 96$, $p = 0.016$). (c) Work sharing on writing documents such as certificates, ($n = 96$, $p = 0.017$). (d) Work sharing on transporting patients. ($n = 96$, $p = 0.043$). P value was calculated by χ2-test. *$p<0.05$, **$p<0.01$.

$p<0.001$) [Fig 4(A)], (10.9% vs. 17.2% vs. 50.0%, $p = 0.006$) [Fig 4(B)], (25.5% vs. 48.3% vs. 75.0%, $p = 0.003$) [Fig 4(C)].

## 4 Discussion

Our study revealed the working hours of hospital pediatricians and OB/GYNs in Tokushima Prefecture, Japan, factors associated with working hours, and characteristics of pediatricians and OB/GYNs who work long hours. Approximately 80% of pediatricians and OB/GYNs working in Tokushima Prefecture, Japan, participated in this study. This study is representative of the Tokushima Prefecture and not Japan; therefore, its results should be used as a guide to conduct further larger studies on pediatricians and OB/GYNs.

Maternal and perinatal mortality rates in Japan are among the lowest worldwide [32]. However, the uneven distribution of pediatricians [26] and OB/GYNs [28] is becoming increasingly apparent. To continuously maintain local medical care systems, including pediatric emergency and perinatal/neonatal care, the results of this study should be used to improve the pediatricians and OB/GYNs' working environments.

### Working hours

In this study, approximately 40% and 10% of pediatricians and OB/GYNs worked for ≥60 h/week and ≥80 h/week, respectively (Table 1). Reports on physicians' working hours vary according to the country where the study was conducted and the categories of physicians surveyed (for example, specialties and residents). However, these results were consistent with

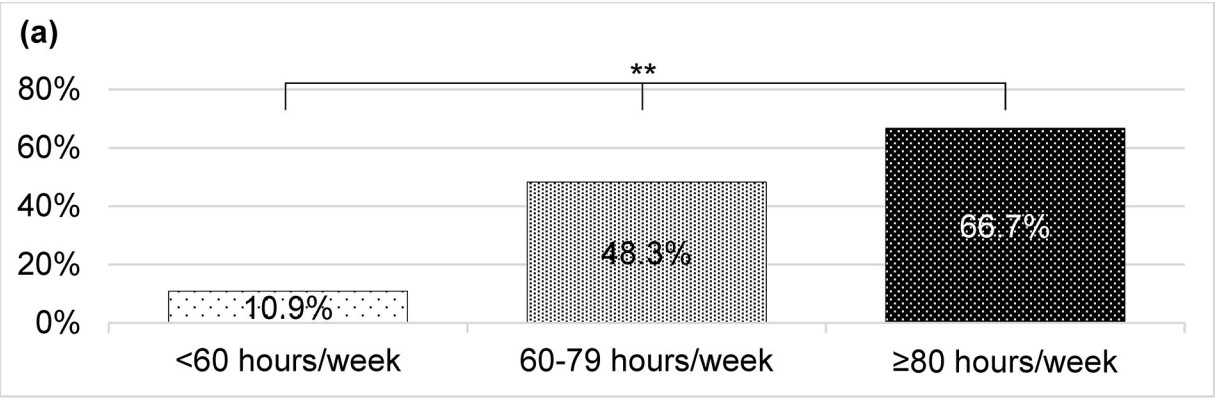

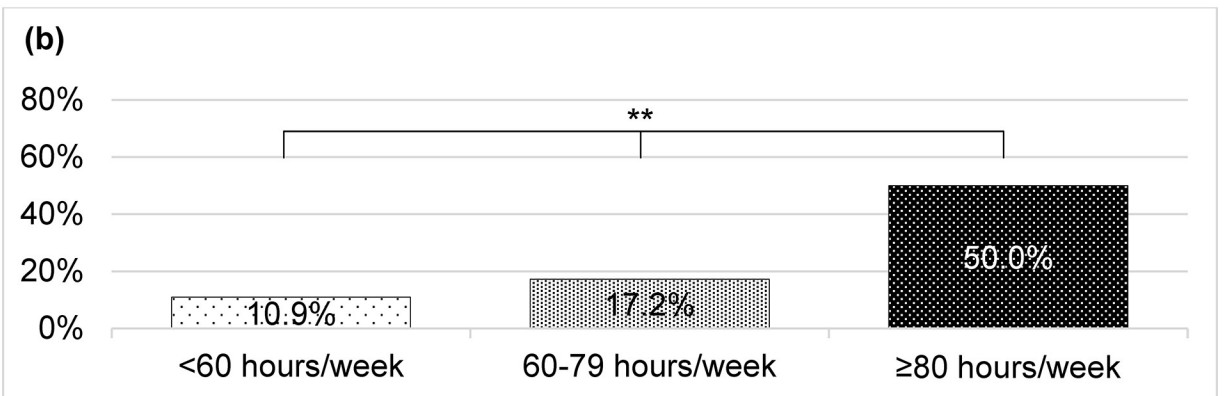

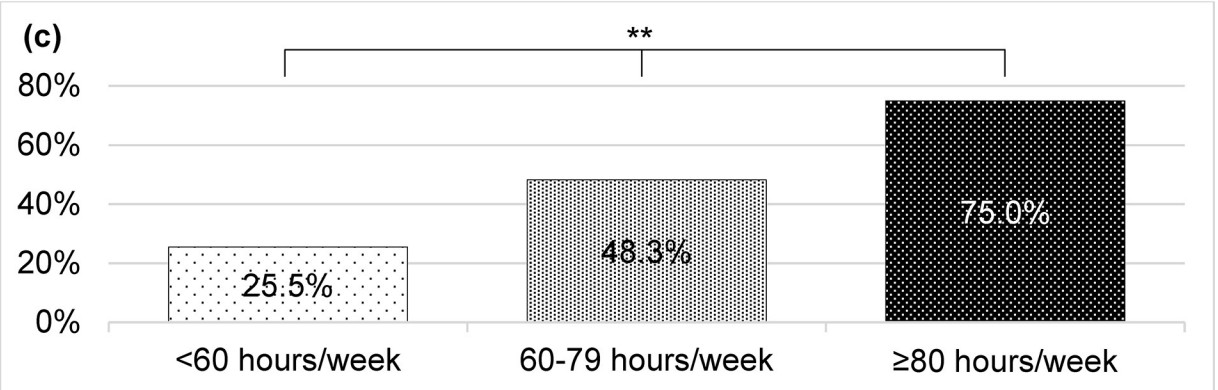

**Fig 4. Percentages of pediatricians and obstetricians/gynecologists working long hours, based on the desired working environment.** (a) Desire to decrease days of night duties ($n$ = 96, $p<0.001$). (b) Desire to decrease days of day-off duties ($n$ = 96, $p$ = 0.006). (c) Desire to decrease overtime work ($n$ = 96, $p$ = 0.003). P value was calculated by χ2-test. $^*p<0.05$, $^{**}p<0.01$.

those of a large-scale study conducted in Japan before the coronavirus disease 2019 (COVID-19) pandemic [16]. In a study conducted in Japan during the COVID-19 pandemic, 51.7% and 14.4% of pediatricians worked for ≥60 h /week and ≥80 h/week, respectively [33]. In another study conducted in Japan after the COVID-19 pandemic, 84% and 47% of Japanese OB/GYNs worked ≥60 h/week and ≥80 h/week, respectively [34]. Compared with previous studies in Japan during or after the COVID-19 pandemic, pediatricians and OB/GYNs worked shorter

hours, which could be due to the recent decline in the birth rate in Tokushima Prefecture compared with that of Japan [35] (Japan: 7.0 vs. Tokushima: 6.3 per 1,000 population, 2019) [36]. Japan's total fertility rate has recently remained flat, and many regions of Japan are expected to continue to experience a declining birthrate [32]. Discussions on the required number of pediatricians and OB/GYNs based on projections of future patients and the number of births are needed.

The COVID-19 pandemic has not necessarily increased working hours [37]. A decrease in the number of pediatric emergency visits to hospitals during the pandemic has been reported [38, 39]. We believe the decline in pediatric and OB/GYN patient numbers may be owing to a decrease in infectious diseases or refrain from visiting clinics owing to the fear of infection. Moreover, the annual number of births in Japan was 840,835 in the pre-pandemic (2019) which decreased to 811,622 and 770,759 during the pandemic (2020 and 2021) [40]. Our study was conducted during the COVID-19 pandemic, and it is possible that fewer labor hours were reported than before. Therefore, checking whether the number of patients and births after the COVID-19 pandemic has recovered to pre-pandemic levels is essential.

Pediatricians and OB/GYNs in their 50s worked ≥60 h/week and 80h/week the most [Fig 1 (A)], with the latter more than the former [Fig 1(B)]. A large survey of physicians in all medical specialties in Japan reported that the adjusted odds ratio for working long hours was significantly lower for those aged ≥40 compared than those aged <30 [22]. We believe certain reasons exist for the long work hours specific to obstetricians and gynecologists. Ishikawa suggested that Japan has increased the management responsibility for middle-aged OB/GYNs in addition to excessive expectations from the public to maintain high-quality medical care [34]. Merlier et al. suggested that French OB/GYNs are at high risk of burnout because of the highly demanding nature of the profession that requires continuous (24-h a day, 7 days a week) care services [41]. Additionally, OB/GYNs are exposed to litigation risks [42, 43]. Therefore, reducing the burden on OB/GYNs, particularly middle-aged OB/GYNs, is essential.

In Japan, the maximum overtime hours in a year for resident doctors and physicians working in emergency departments will be limited to 1,860 h (equivalent to an 80-h work/week) and 960 h (equivalent to a 60-h work/week) for others, based on the Labor Standards Act starting in FY2024 [31]. Assuming legal working hours of 40 h/week and 50 weeks/year, the 60-h and 80-h work/week corresponds to 1,000 h/year and 2,000 h/year of overtime excluding legal working hours, respectively. Pediatricians and OB/GYNs who work ≥80 h/week must work <80 h/week at the start of FY2024. Until FY2024, the law in Japan had not set uniform standards for physician's overtime hours. The items identified in this study (factors associated with long working hours and how to reduce them) should be used to improve hospital pediatricians and OB/GYNs' working environments.

## Long working hours

In this study, weekly hours of work by pediatricians and OB/GYNs were significantly associated with working at night and day-off duties and the number of medical institutions being worked at concurrently (Table 4). Moreover, the association was more related to working at night and day-off duties than the number of medical institutions being worked at concurrently (Table 4). Many physicians who work long hours want to reduce the number of days of night and day-off duties [Fig 4(A) and 4(B)]. We suggest a shortage of pediatricians and OB/GYNs to cover night and day-off duties in hospitals due to the uneven distribution of pediatricians and OB/GYNs and working hours, and the number of night and day-off duties has increased to compensate for it. Therefore, it is necessary to strengthen the recruitment and retention of pediatricians and OB/GYNs at medical institutions in the future.

However, previous studies have reported that many OB/GYNs perceive their salaries as low because of the nature of their work [41]. The average annual salary paid to hospital physicians in Japan by their hospitals is lower than that of medical practitioners [44]. An independent association was observed between long working hours and high annual income for pediatricians [33]; hence, due to financial circumstances, physicians working at the hospital may possibly be working at other medical institutions.

From FY2024, the maximum annual working hour limit for physicians will be introduced [31]. The shortage of pediatricians and OB/GYNs in hospitals may make maintaining local medical care systems difficult. From a long-term perspective, in addition to increasing the number of pediatric and OB/GYN residents, consolidating hospitals that provide pediatric emergency and perinatal/neonatal care may be necessary.

## Efforts to reduce long working hours

In this study, working hours were significantly shorter in the "multiple attending physician system" than in the "primary attending physician system," and in the "on-call system" than in the "shift work system" ($p < 0.05$) (Fig 2). We suggest that increasing the number of pediatricians and OB/GYNs per medical institution is necessary to achieve this working status. However, in previous studies, a higher number of on-call schedules per month resulted in a higher percentage of burnout, whereas a higher number of on-call physicians resulted in a lower percentage of burnout [41]. Hence, we believe that the fundamental solution to this problem is increasing the number of physicians per medical institution.

In 2012, the World Health Organization recommended that task-shifting from physicians to nonphysicians is effective in protecting maternal and newborn health [45]. The Japanese Health Ministry recommends task-shifting and task-sharing [46]. For example, some tasks have been shifted from physicians to non-physicians, such as clinical radiologists, clinical laboratory technicians, and clinical engineers. Work-style reforms for physicians in the hospitals which are the primary physician team system, and the cross-functional teams between hospital staff such as physicians, pharmacists, and nurses, are promoted. In a study of pediatricians [47] and OB/GYNs [48] in Japan regarding task shifts, approximately 60% of pediatricians and 50% of OB/GYNs favored task shifts. In the same study, pediatricians and OB/GYNs indicated that they could reduce their working hours by approximately 2 h/day. In contrast, some participants objected to the promotion of task shift in the specific tasks of pediatrics and OB/GYN, such as "Venous blood sampling (newborn/infant)" and "Fetal echogram at prenatal check-up." In this study, the proportion of physicians who worked longer hours (60 h/week or 80 h/week) may have declined significantly during the Task Shift ($p < 0.05$) (Fig 3). Notably, the task shift duties surveyed in this study are not items that would be answered as objectionable in the previous studies [47, 48]. Therefore, task shifts for pediatricians and OB/GYNs are essential to improve their working environment.

## Desired working status

Our study found that pediatricians and OB/GYNs at hospitals with long working hours (over 80 h/week) were less willing to work longer hours and decreased their night and day-off duties than those working fewer hours (Fig 4). Particularly, pediatricians working >60 h/week desired a decrease in night duties. Night duties not only adversely affect physicians' quality of life [41] but are associated with short sleep duration and sleep disorders [49], which are associated with burnout among physicians [50]. Additionally, long working hours without breaks threaten the safety of medical care [51, 52]. Repeatedly, the fundamental solutions would require more young physicians in pediatrics and OB/GYN at hospitals and the consolidation of hospitals that provide pediatric emergency and perinatal/neonatal care.

Working hours and environment may be related to physicians' careers and job satisfaction [53]. High career and job satisfaction negatively affect burnout [13, 54]. Additionally, high job satisfaction may keep physicians in the hospital. In addition to actively recruiting new physicians at the hospital level, efforts must be made to reduce the burden on physicians. A large-scale study on the details of the desired working status should be conducted in the future.

## Limitations

This study had some limitations. First, the sample size was small because it targeted pediatricians and OB/GYNs working at hospitals in Tokushima Prefecture, Japan. Because the analysis was conducted collectively by professionals involved in perinatal and neonatal care, the specific trends in hospital installations and departments are unclear. Second, this was a cross-sectional study; therefore, causal relationships were unclear. Third, selection and recall bias are possible in this study. Approximately 20% of the participants did not respond to this study, which may have included busy pediatricians and OB/GYNs. The number of working hours per week was based on self-reported data. However, this is a common method used in epidemiological studies of physicians' working hours because the responses must be simple. Fourth, some items were not included in the questionnaire but were reportedly related to long working hours. Items related to the type of hospital where they worked, details of their job, salary, and family and living arrangements were not included in the questionnaire. Particularly, we did not ask the OB/GYNs about combined obstetric and surgical activities. Previous studies have reported that approximately 60% of the physicians did not combine the activities [41]. Additionally, some reports suggest a relationship between gender roles, child-rearing, and working hours [55]. In the future, considering conducting a larger-scale study that includes these items while reducing the burden on respondents is necessary.

## 5 Conclusions

This study revealed that approximately 40% and 10% of hospital pediatricians and OB/GYNs in Tokushima, Japan, work $\geq 60$ h/week and $\geq 80$ h/week, respectively. Their weekly working hours were associated with working at night duties and day-off duties, number of working medical institutions concurrently. Physicians who worked long hours identified issues with sharing work with medical or non-medical workers and desired a reduction in the number of night and day-off duties and working hours. Our findings provide insights into improving the working environments of pediatricians and OB/GYNs. Hence, conducting a detailed and large-scale study of the working environments of pediatricians and OB/GYNs in the future is necessary.

## Supporting information

**S1 File.**
(DOCX)

**S2 File.**
(ZIP)

## Acknowledgments

We wish to express our gratitude to the hospital physicians and hospitals who permitted us to undergo this survey and to Mrs. Shinobu Isomoto and Sanae Mori (Department of Public Health, Faculty of Medicine, Tokushima University) for their assistance with this study.

## Author Contributions

**Conceptualization:** Kazumi Nakagawa, Takeshi Iwasa, Hisayoshi Morioka.

**Data curation:** Takeshi Iwasa, Maki Urushihara, Hisayoshi Morioka.

**Formal analysis:** Mai Nakagawa, Kazumi Nakagawa, Koga Nakai, Ayumu Tominaga, Ichiro Hashimoto, Hisayoshi Morioka.

**Investigation:** Takeshi Iwasa, Maki Urushihara, Hisayoshi Morioka.

**Methodology:** Hisayoshi Morioka.

**Project administration:** Hisayoshi Morioka.

**Writing – original draft:** Mai Nakagawa, Kazumi Nakagawa, Koga Nakai, Ayumu Tominaga, Hisayoshi Morioka.

**Writing – review & editing:** Mai Nakagawa, Kazumi Nakagawa, Yoshiro Mori, Takeshi Iwasa, Maki Urushihara, Ichiro Hashimoto, Hisayoshi Morioka.

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
