## [Decision Letter · Decision Letter 0]

7 Aug 2024

PONE-D-24-08882Characteristics of hospital pediatricians and obstetricians/gynecologists working long hours in Tokushima, Japan: A cross-sectional studyPLOS ONE

Dear Dr. MORIOKA,

Thank you for submitting your manuscript to PLOS ONE. After careful consideration, we feel that it has merit but does not fully meet PLOS ONE’s publication criteria as it currently stands. Therefore, we invite you to submit a revised version of the manuscript that addresses the points raised during the review process.

We look forward to receiving your revised manuscript.

Kind regards,

Jorge Cervantes

Academic Editor

PLOS ONE

5. We note that you have referenced (Kivimäki M, Jokela M, Nyberg ST, Singh-Manoux A, Fransson EI, Alfredsson L, et al. Long working hours and risk of coronary heart disease and stroke: a systematic review and meta-analysis of published and unpublished data for 603,838 individuals. Lancet. 2015;386:1739-1746.) which has currently not yet been accepted for publication. Please remove this from your References and amend this to state in the body of your manuscript: (ie “Bewick et al. [Unpublished]”) as detailed online in our guide for authors

Reviewers' comments:

Reviewer's Responses to Questions

**Comments to the Author**

1. Is the manuscript technically sound, and do the data support the conclusions?

Reviewer #1: Partly

Reviewer #2: Yes

2. Has the statistical analysis been performed appropriately and rigorously? 

Reviewer #1: No

Reviewer #2: Yes

3. Have the authors made all data underlying the findings in their manuscript fully available?

Reviewer #1: Yes

Reviewer #2: Yes

4. Is the manuscript presented in an intelligible fashion and written in standard English?

Reviewer #1: Yes

Reviewer #2: Yes

5. Review Comments to the Author

Reviewer #1: Authors undergo cross-section surveying study on pediatrics, Obstetrics, and Gynecologists in a hospital in Tokushima prefecture however the study has many limitations. The authors have highlighted most of the limitations here. However, much information should be revised to accept this manuscript.

The introduction lacks much information about the main problem (Working long hours) reasons and why your study might be the initial step to solve part of the problem.

The authors need to mention the physician's normal standardized hours required from each institution to understand the abnormal or overtime hours.

Did the normal standardized hours for work per week differ from one institution and another? Authors should confirm this point in their analysis. did the paid leave hours differ? What is the normal or maximum paid leave days annually? It is difficult to understand the problem WITHOUT mentioning the main standardized normal values.

I added a review note file for all my comments.

Reviewer #2: Technically sound manuscript reflecting original research. The data provided suppors the conclusions made in this manuscript. This study seems relevant to the changing legal landscape regarding working hour in Japan.

6. PLOS authors have the option to publish the peer review history of their article (what does this mean?). If published, this will include your full peer review and any attached files.

Reviewer #1: **Yes: **Ahmed M. Abdou South Valley University, Egypt

Reviewer #2: No

---

## [Author Response · Author response to Decision Letter 0]

20 Sep 2024

Response to Reviewing

We greatly appreciate the in-depth and precise review of our manuscript by the reviewers. The instructive comments by the reviewers and editor were indeed helpful, and we have modified the manuscript based on the suggestions provided. The revisions have been listed below.

Author’s Response to Comment of Reviewers

Major comments

Keywords:

Reviewer Comment: Please remove any abbreviations from here. It will be abbreviated in the text.

Response: Following the reviewer’s advice, we have deleted the abbreviations from the keywords. 

(see Marked version page 6, line 72)

The introduction:

Reviewer Comment: The introduction lacks much information about the main problem (Working long hours) reasons and why your study might be the initial step to solve part of the problem. The authors need to mention the physician's normal standardized hours required from each institution to understand the abnormal or overtime hours.

Response: Following the reviewer’s advice, we have added the information about normal standardized and overtime hours for workers including physicians.

(see Marked version page 7, lines 77-84)

Reviewer Comment: Line 103: Japan Labor standards will be revised from FY 2024. Please mention the previous standards or highlight them so that the readers can understand at which points the update will be done.

Response: Following the reviewer’s advice, we have included the information on previous Japanese Labor standards for overtime limitation.

(see Marked version page 9, lines 122-126)

Reviewer Comment: Lines 83-86: Why did authors only on a study from Taiwan? More Japanese studies here will be more useful to be added here.

Response: Following the reviewer’s advice, we have added information on previous Japanese studies about physician’s working hours.

(see Marked version page 8, lines 98-102, and page 8, lines 105-107)

Material and methods:

Reviewer Comment: I do request to add an ethical statement because the current way is confusing and unclear. Please write a paragraph, that clarifies “the ethics of the research” at the beginning of the material and methods section.

Response: Following the reviewer’s advice, we have moved the ethical statement to the beginning of the material and methods section. and have created a paragraph with the heading “Ethics of the research.”

(see Marked version page 10, line 140-page 11, line 146)

Reviewer Comment: Line 130: Can you add a copy template from this questionnaire in the supplementary information along with this manuscript? It is not confidential information.

Response: For this study, we prepared the questionnaire in Japanese. Following the reviewer’s advice, we have added a copy of the main questions translated into English as supporting information 1.

(see Marked version page 15, lines 185)

Reviewer Comment: Did permission needed from the administrative office in the hospital to provide the information? please provide this information.

Response: No hospital’s administrative office permission was needed to conduct this survey. We have included this information in the manuscript.

(see Marked version page 12, lines 169-170)

Reviewer Comment: Line 140: the number of participants in this study was low. It might be a big limitation in this study.

Response: We also considered the small sample size as a limitation of this study. The reason for this is the small number of pediatricians and obstetricians working in hospitals in Tokushima Prefecture. This point has been mentioned at the beginning of the limitation section of this manuscript. 

(see Marked version page 44, lines 558-562)

Reviewer Comment: Line 153: What did the authors think that sex might affect the activity of work and long-hour work in this manuscript?

Response: We have provided the results of our analysis for sex difference in Table 2. The analysis showed no significant difference in weekly working hours by sex. 

(see Marked version page 25, Table 2)

Reviewer Comment: Authors should add the factors affecting long hours work surveyed by the recently reported studies as https://doi.org/10.1186/s12889-023-17531-5 here and in the discussion as well.

Response: Following the reviewer’s advice, we have revised our discussion regarding the study you presented.

(see Marked version page 38, lines 448-452)

Reviewer Comment: Line 188: what is the reference of the chosen hours surveyed in this study? Please mention the reference if possible or explain why it if it a new criterion.

Response: Following the reviewer’s advice, we have added the information on the criteria used for the analysis.

(see Marked version page 17, lines 224-230)

Statistical analysis:

Reviewer Comment: Please mention here that each statistical test that was used was mentioned in each figure legend and mention it in each figure legend also.

Response: Following the reviewer's advice, we have included notes the statistical tests used in the legend of each figure. Moreover we have included a statement to that effect in the manuscript.

(see Marked version page 19, lines 259)

Reviewer Comment: Can authors reconfirm the statistical analysis by test rather than a Chi-square test? Would be better.

Response: Following the reviewer's advice, we have modified our statistical analysis methodology for factors associated with longer work hours. Since the original data for working hours were continuous variables, we reanalyzed them using Pearson correlation analysis and hierarchical regression analysis (Table 3and Table 4). Accordingly, we have changed the description of the Abstract, Methods and Results.

(see Marked version page 4, line 50-page 5, line 58, page 18, line 241-page 19, line 252, and page 26, line 318-page 30, line 345)

Results:

Reviewer Comment: Rewrite the result writing to explain the analysis done illustrating the significant elevation and decrease rather than writing Figure 1 or 2 shows….. the data explanation should be explained in a more scientific concise way. The current way just shows what is in the figure.

Response: Following the reviewer's advice, we have included a description of the of the results of the detailed analysis.

(see Marked version page 20, line 268-page 34, line 390)

Reviewer Comment: Lines 211: Participants work over 80 hours were very low. The titles and tasks given to these people might be important. Did the authors have any information on this?

Response: More than half of the physicians in this survey who worked more than 80 hours per week were aged ≥40 years. In addition, more physicians working ≥80 hours per week were in obstetrics and gynecology than in pediatrics. A previous study of hospital physicians of all medical specialties in Japan reported lower adjusted odds ratios for working longer hours (>1,860 h/year) for those aged ≥40 than for those aged <40 years [17]. Therefore, we believe that the management responsibility for middle-aged OB/GYNs and the lack of young OB/GYNs are responsible for the long working hours. We have addressed this point in our paper.

[17] Koike S, Wada H, Ohde S, Ide H, Taneda K, Tanigawa T. Working hours of full-time hospital physicians in Japan: a cross-sectional nationwide survey. BMC Public Health. 2024;24:164.

(see Marked version page 38, lines 448-452)

Reviewer Comment: Line 231, 244, and 260: Delete the words ‘ Figure XX shows’ as the text information after it already referred to with a Figure number in brackets at the end of the paragraph.

Response: Following the reviewer's advice, we have removed the sentence you point out.

Reviewer Comment: For figures in the text, Please follow the journal guidelines in referring to certain results; for example Figure 2-b may not be correct.

Response: Following the reviewer’s advice and the journal guidelines, we have corrected the figure numbering.

Figures:

Response: Following the reviewer’s advice, we have reviewed all the Figures.

Tables:

Response: Following the reviewer's advice, we have revised Table 1. We have also removed Table 2 and added Tables 2-4 to show the results of the new analysis.

Discussion

Reviewer Comment: The discussion need much literature concerned about factors affecting long-hour work, reasons for overtime work, and what is the recommended solution to overcome the problem of burnout.

Response: Following the reviewer’s advice, we have included sentences linked to factors affecting long-hour work.

(see Marked version page 39, line 474-page 40, line 484, and page 40, lines 488-489)

Reviewer Comment: Line 295: How did the authors think that Coronavirus pandemic can affect the pediatrics, Obs/Gyn working hours despite the difference in the specialty from those concerned with the pandemic patients? The explanation should be more clear here.

Response: Following the reviewer’s advice, we have added notes on the impact of the COVID-19 pandemic on pediatric and obstetric care. 

(see Marked version page 37, lines 436-442)

Reviewer Comment: Line 326-327: I think the correspondence is 2,000 to 3,000 hours not 1,000 to 2,000. Please confirm.

Response: Notably, 60-h/week and 80-h/week work hours are equivalent to 1,000 h/year and 2,000 h/year, respectively, as overtime excluding legal working hours. We have added a note to the manuscript that it is overtime, excluding legal working hours.

(see Marked version page 39, line 466)

Reviewer Comment: Line 316-319: Please mention that this information is from a study done in France.

Response: Following the reviewer’s advice, we have added that the survey is for French OB/GYNs.

(see Marked version page 38, line 455)

Reviewer Comment: Line 323-324: These hours time is per month or semester or what?

Response: Following the reviewer’s advice, we have added that these overtime hours are per year.

(see Marked version page 38, line 460)

Reviewer Comment: Line 325: if possible, mention the overtime regulation stated by the Ministry of Labor, health in Japan for better clarification.

Response: Following the reviewer’s advice, we have added information on Japanese overtime regulations prior to FY 2024.

(see Marked version page 39, lines 467-468)

Reviewer Comment: Line 343: This study was done in France. Please find out more studies done in Japan. The salaries of physicians in Japan are not low to affect working hours. Please explain this point.

Response: This survey covered national, prefectural and municipal, and public hospitals (including Japanese Red Cross, National Welfare Federation of Agricultural Cooperatives and Japan Community Health Care Organization). According to statistics from the Ministry of Health, Labor and Welfare [44], the average salaries of physicians working at national, prefectural and municipal, and public hospitals are 106,478 U.S. dollars (USD) (14,101 thousand Japanese yen (JPY)), 109,922 USD (14,557 thousand JPY), and 109,620 USD (14,517 thousand JPY) in FY2022. On the other hand, the average salaries of medical corporations and hospitals managed privately, which are the recipients of human resources support, are 113,146 USD (14,984 thousand JPY) and 128,588 USD (17,029 thousand JPY). The average salaries of clinics (with beds and without beds) operated by medical corporations, which may also receive human resources support, are 259,586 USD (34,377 thousand JPY) and 194,683 USD (25,782 thousand JPY). This average salary amount is paid by the physician's affiliated facility and includes overtime, dependent care, and commuting allowances. In Japan, doctors' salaries are not considerably different by clinical department. However, the amount of salary may vary depending on the specialty of the clinic. Therefore, we believe that physicians working in national, prefectural and municipal, and public hospitals are paid less than physicians working in other health care institutions. Hence, we considered that this may be one of the motivations for those physicians to work at medical institutions other than their own. We have added to the manuscript that the average annual salary paid by hospitals to physicians who work at medical institutions in Japan is lower than that of medical practitioners. (1 dollar=132.43 JPY, Yen-dollar annual average in 2022)

[44] Ministry of health, labor and welfare. The 24th of Economic Conditions in Healthcare (Survey on Healthcare Facilities), Japan. 2023. [cited 2024 Aug 22]. Available from: https://www.mhlw.go.jp/bunya/iryouhoken/database/zenpan/jittaityousa/dl/24_houkoku_iryoukikan.pdf (in Japanese).

(see Marked version page 40, lines 491-493)

Reviewer Comment: Line 346: Please state Japan's regulations for working in more than one institution in response to working hours. This information is very important and might affect the analysis.

Response: The survey covered three types of medical institutions: national, prefectural and municipal, and public hospitals (including the Japanese Red Cross, National Welfare Federation of Agricultural Cooperatives, and Japan Community Health Care Organization). Physicians working in such hospitals are often subject to local or national civil service regulations. Physicians who wish to work concurrently may do so with the permission of the hospital director. Cases in which permission for dual employment may not be granted are when it interferes with the main business, when self-employed, or when serving as an executive officer of a for-profit company. Therefore, physicians may work at other medical institutions with the permission of the hospital director to the extent that it does not interfere with their main job.

Reviewer Comment: Line 367: Please state more details of task shifting and sharing from reference 36.

Response: Following the reviewer’s advice, we have added an example of task shifting and sharing in Japan.

(see Marked version page 42, lines 517-521)

Reviewer Comment: Line 377: information come from 2 reported papers, please check grammer here is supposed to be ‘Studies’ not single study.

Response: We have made this correction in accordance with your suggestion.

(see Marked version page 42, line 532)

Ethics approval and consent to participate

Reviewer Comment: Better to transfer this section to the beginning of the Material and methods section.

Response: Following the reviewer’s advice, we have transferred the description of the ethical considerations to the beginning of the “Methods” section.

(see Marked version page 10, line 140-page 11, 146)

Funding

Reviewer Comment: Please provide both Grant name and number here.

Response: Following the reviewer’s advice, we have added the name of the grant. However, no number was assigned to this grant.

(see Marked version page 48, lines 618-619)

Acknowledgments

Reviewer Comment: Please add acknowledgment to the medical institutions and hospitals physicans or directors which the participants belong for permitting you to undergo this survey.

Response: Following the reviewer’s advice, we have acknowledged the hospital physicians and hospitals who permitted us to undertake in this survey.

(see Marked version page 49, lines 631-632)

---

## [Editor Report · Decision Letter 1]

24 Sep 2024

Characteristics of hospital pediatricians and obstetricians/gynecologists working long hours in Tokushima, Japan: A cross-sectional study

PONE-D-24-08882R1

Dear Dr. MORIOKA,

We’re pleased to inform you that your manuscript has been judged scientifically suitable for publication and will be formally accepted for publication once it meets all outstanding technical requirements.

Kind regards,

Jorge Cervantes

Academic Editor

PLOS ONE
---

## [Editor Report · Acceptance letter]

7 Nov 2024

PONE-D-24-08882R1 

PLOS ONE

Dear Dr. Morioka, 

I'm pleased to inform you that your manuscript has been deemed suitable for publication in PLOS ONE. Congratulations! Your manuscript is now being handed over to our production team.

Kind regards, 

on behalf of

Dr. Jorge Cervantes 

Academic Editor

PLOS ONE